# Comprehensive Genome-Wide Analyses of Poplar R2R3-MYB Transcription Factors and Tissue-Specific Expression Patterns under Drought Stress

**DOI:** 10.3390/ijms24065389

**Published:** 2023-03-11

**Authors:** Xueli Zhang, Haoran Wang, Ying Chen, Minren Huang, Sheng Zhu

**Affiliations:** 1Key Laboratory of Forest Genetics and Biotechnology, Ministry of Education of China, Co-Innovation Center for the Sustainable Forestry in Southern China, Nanjing Forestry University, Nanjing 210037, China; 2College of Biology and the Environment, Nanjing Forestry University, Nanjing 210037, China

**Keywords:** poplar, R2R3-MYB transcription factors, collinear analysis, drought-responsive expression pattern, tissue-specific

## Abstract

R2R3-type MYB transcription factors are implicated in drought stress, which is a primary factor limiting the growth and development of woody plants. The identification of *R2R3-MYB* genes in the *Populus trichocarpa* genome has been previously reported. Nevertheless, the diversity and complexity of the conserved domain of the *MYB* gene caused inconsistencies in these identification results. There is still a lack of drought-responsive expression patterns and functional studies of R2R3-MYB transcription factors in *Populus* species. In this study, we identified a total of 210 *R2R3-MYB* genes in the *P. trichocarpa* genome, of which 207 genes were unevenly distributed across all 19 chromosomes. These poplar *R2R3-MYB* genes were phylogenetically divided into 23 subgroups. Collinear analysis demonstrated that the poplar *R2R3-MYB* genes underwent rapid expansion and that whole-genome duplication events were a dominant factor in the process of rapid gene expansion. Subcellular localization assays indicated that poplar R2R3-MYB TFs mainly played a transcriptional regulatory role in the nucleus. Ten *R2R3-MYB* genes were cloned from *P. deltoides* × *P. euramericana* cv. Nanlin895, and their expression patterns were tissue-specific. A majority of the genes showed similar drought-responsive expression patterns in two out of three tissues. This study provides a valid cue for further functional characterization of drought-responsive *R2R3-MYB* genes in poplar and provides support for the development of new poplar genotypes with elevated drought tolerance.

## 1. Introduction

Over the years, global climate change has led to an increase in the frequency and spatial extent of droughts, severely reducing the stress resistance and resilience of perennial woody plants and even jeopardizing the functioning of forest ecosystems [1,2]. Poplar is a primary tree species that is distributed widely in Northern Hemisphere regions, especially in dry northern and northwestern China [3,4]. Poplar is also a model woody plant for studying abiotic stress response mechanisms, owing to its characteristics, such as a relatively small genome, high genetic diversity, relative ease of vegetative propagation and genetic transformation, short rotation cycle, and fast growth rate [5,6]. However, very little is known about the molecular regulatory mechanism of poplar under soil water deficit.

Transcription factors (e.g., MYB, NAC, and WRKY) are essential components of the transcriptional regulatory network in plant species under abiotic stresses [7]. R2R3-type MYB proteins are the largest MYB subfamily transcription factors related to abiotic stress responses (e.g., drought, dehydration, heat, and salinity) in plants [8,9]. Many studies have documented that plant *R2R3-MYB* genes might affect drought resistance by thickening leaf cuticular waxes, controlling stomatal aperture, and regulating the expression of ABA signaling pathway genes [10,11]. For example, *AtMYB94* and *AtMYB96* can influence the expression of wax biosynthesis-related genes (e.g., *WSD1* and *KCS2*) by specifically binding to the MYB binding consensus sequence (MBS) in their promoter region [12,13]. *AtMYB44* positively regulated ABA to induce stomatal closure and suppressed the expression of the protein phosphatase 2C (PP2C) genes (e.g., *ABI1*, *ABI2*) in *Arabidopsis* [14,15]. *PtrMYB94* (Potri.017G082500) can regulate the expression of ABA and drought stress-related genes (e.g., *ABA1*, *DREB2B* and *P5CS2*) under drought stress [16]. *PtoMYB170* plays a role in lignin deposition during poplar wood formation and can also reduce the rate of water evaporation by closing foliar stomata [17]. *PtoMYB142* can directly bind to the promoters of wax biosynthesis genes (e.g., *CER4* and *KCS6*) and induce their expression, resulting in increased wax accumulation in poplar leaves to adapt to water scarcity [18].

Accurate identification of *R2R3-MYB* genes throughout the genome is required to comprehensively analyze their functional roles in various abiotic stress responses. With the increasing number of flowering plant genomes being published, genome-wide identification of the *R2R3-MYB* gene family has been reported in many dicotyledonous species, such as *Arabidopsis thaliana* [19], *Vitis vinifera* [20], *Eucalyptus grandis* [21], *Capsicum annuum* [22], *Pinus massoniana* [23] and *Lycium barbarum* [24]. Genome-wide identification of the *R2R3-MYB* genes in the *Populus trichocarpa* genome (v3.0) has also been reported [25,26,27,28]. These studies identified that the total number of *P. trichocarpa R2R3-MYB* genes ranged from 191 to 207. The inconsistencies between these resulting *R2R3-MYB* identifications are partly due to two adjacent conserved MYB repeats at an N-terminal region and a hypervariable regulatory region at a C-terminus [11]. However, a few important *R2R3-MYB* genes in poplar might be missed owing to the inconsistencies.

Here, we identified a total of 210 *R2R3-MYB* genes in the entire *P. trichocarpa* genome. We performed a comprehensive bioinformatic analysis of these *P. trichocarpa R2R3-MYB* genes, including their physicochemical properties, chromosomal distribution, *cis*-acting element and subcellular location prediction, phylogenetic and collinearity analyses, and RNA-seq profiling of *R2R3-MYB* genes in three distinct tissues under drought. A total of 10 *R2R3-MYB* genes were cloned from the poplar genotype NL895 (*P. deltoides* × *P. euramericana* cv. Nanlin895), and their tissue-specific and drought-responsive expression patterns were analyzed via quantitative RT-PR (qPCR). In addition, protein subcellular localization analysis indicated that both PdMYB2R032 and PdMYB2R151 proteins were localized in the nucleus of NL895 protoplasts and coincided with their prediction results. These results provide an important clue for screening drought-related *R2R3-MYB* genes in poplar and for further studying their roles in poplar under drought stress.

## 2. Results

### 2.1. Identification and Characterization of Poplar R2R3-MYB Gene Family Members and Their Chromosomal Distribution

According to the hidden Markov model (HMM) file and the structural characteristics of *R2R3-MYB*, we identified members of the poplar *R2R3-MYB* gene family. In this research, we identified a total of 210 R2R3-MYB transcription factors (TFs) comprising an adjacent MYB repeat pair from the *P. trichocarpa* genome (v3.0) (Appendix A). These R2R3-MYB TFs were compared with those of four previously published papers (Figure 1) [25,26,27,28]. The union set of all *P. trichocarpa* R2R3-MYB TFs reported previously in the four papers comprised 211 protein-coding genes, only one (Potri.015G143400.1) of which was not included in the 210 *R2R3-MYB* genes that we identified. The Potri.015G143400.1 gene was identified as an *R2R3-MYB* gene only in the results of Chai et al. [25]. By examining the number of MYB repeats on the PFAM, CDD, and SMART web services, the Potri.015G143400.1 protein was found to have only a repeat of the MYB domain and hence belonged to the *1R-MYB* gene family. In addition, 189 out of the 210 *R2R3-MYB* genes had an intersection between our results and four previous genome-wide identification results (Figure 1). The remaining 21 poplar R2R3-MYB proteins were not consistent across any of the five resulting sets and were searched manually for two neighboring MYB repeats at their N-terminal region. Thus, the putative 210 *R2R3-MYB* genes identified in this study should belong to the *R2R3-MYB* gene family in *P. trichocarpa*. In addition, these remaining genes might be arisen by whole-genome duplication (WGD) and dispersed duplication.

The location information of these *R2R3-MYB* genes on poplar chromosomes was obtained from the *P. trichocarpa* genome annotation file in GFF/GTF format. Only three genes (Potri.t011400.1, Potri.T125000.1, and Potri.T144800.1) among all 210 *R2R3-MYB* genes were not located on all 19 chromosomes in *P. trichocarpa* and were renamed *PtrMYB2R208*, *PtrMYB2R209*, and *PtrMYB2R210*, respectively. These three genes are located on scaffolds (Appendix A). The other 207 *R2R3-MYB* genes on the 19 *P. trichocarpa* chromosomes were separately renamed *PtrMYB2R001* to *PtrMYB2R207* on the basis of their chromosomal position (Figure 2). As the two with the most *R2R3-MYB* genes, *P. trichocarpa* chromosome 1 (Chr01) and chromosome 17 (Chr17) had 23 (*PtrMYB2R001*-*PtrMYB2R023*) and 15 (*PtrMYB2R173*-*PtrMYB2R187*) *PtrMYB2R* genes, respectively. Only two genes (*PtrMYB2R171* and *PtrMYB2R172*) were located on *P. trichocarpa* Chr16, which had the lowest number of *R2R3-MYB* genes. The *R2R3-MYB* genes were relatively uniformly distributed over the remaining 16 chromosomes, with nearly 10 *R2R3-MYB* genes and almost one gene per 2 Mb.

### 2.2. Physicochemical Property Analysis of Poplar R2R3-MYB Proteins

To some extent, the physiological and biochemical properties of proteins can provide a reliable theoretical basis for functional research on R2R3-MYB TFs. Therefore, we predicted the physicochemical properties of these poplar R2R3-MYB family members, as shown in Appendix A. The protein lengths of PtrMYB2Rs ranged from 162 (PtrMYB2R084) to 1070 (PtrMYB2R142) amino acids, with an average length of 335 amino acids. The molecular weight of the proteins ranged from 18.47 kDa (PtrMYB2R084) to 119.55 kDa (PtrMYB2R142), with a mean of 37.67 kDa. In addition, the isoelectric point (pI) of poplar R2R3-MYB proteins ranged from 4.71 (PtrMYB2R150) to 9.79 (PtrMYB2R179), and the aliphatic index ranged from 50.54 (PtrMYB2R094) to 83.74 (PtrMYB2R049). The aliphatic index determines thermostability of globular proteins, indicating their high thermal stability and flexibility. Among the poplar R2R3-MYB family, only six were stable proteins (PtrMYB2R008, PtrMYB2R018, PtrMYB2R023, PtrMYB2R130, PtrMYB2R182 and PtrMYB2R183), and their instability index was less than 40. The remaining 204 R2R3-MYB proteins were unstable (instability index > 40) and they were randomly distributed in each subgroup. This instability of R2R3-MYBs in poplar is similar in other transcription factor families, such as TIFY [29], ERF [30], and E2F-DP [31]. In addition, the grand average of hydropathicity (GRAVY) of all proteins in this family was less than 0, illustrating that these 210 R2R3-MYB proteins might be all hydrophilic proteins. These results indicated that the great differences in the properties of poplar R2R3-MYB proteins might be implicated in plant biological and abiotic stresses, and these proteins might play important roles in plant growth and development [11].

### 2.3. Analysis of Cis-Acting Elements in R2R3-MYB Genes

Ci*s*-acting elements are closely related to specific biological functions and may participate in the regulation of gene expression. Hence, we analyzed the *cis*-acting elements in the promoter upstream of all the *PtrMYB2R* genes (Figure 3). The promoter region of the *PtrMYB2R* genes had the highest number of *cis*-elements involved in abiotic stress, including MYB, MYC, STRE (stress response elements), ABRE (ABA-responsive element), TATA-box, G-box, W-box, and MBS (MYB-binding sites). In addition to STRE, which is involved in osmotic stress and heat shock, other *cis*-acting elements can bind to transcription factors (e.g., MYB, WRKY, and DREB) to regulate the biological process of the plant response to drought stress [32,33]. We also found that there were many light response elements (Box-4) and oxidation defense elements (e.g., ARE (antioxidant response element), CGTCA motif, and AAGAA motif) [34,35]. In addition, there were some *cis*-elements related to plant hormone biosynthesis, which were mainly involved in the biosynthesis of MeJA (CAAT-box, CGTCA-motif, and TGACG-motif), ethylene (ERE, ethylene response element), and salicylic acid (TCA element, salicylic acid-responsive elements) [34,36]. Thus, the number of *cis*-acting elements in the promoter region of the *R2R3-MYB* genes was large and rich in variety (Appendix A).

### 2.4. Phylogenetic Analysis of the Poplar R2R3-MYB Gene Family

To better understand the evolutionary relationship between poplar *R2R3-MYB* genes, we constructed an unrooted ML (maximum likelihood) tree using all the poplar R2R3-MYB proteins. More than 90% of branches in the ML tree had a bootstrapping value of more than 80%, indicating the reliability of subgroup classification (Figure 4). All 210 R2R3-MYB proteins in the entire *P. trichocarpa* genome were distinctly divided into 23 subgroups, the S16 subgroup of which had only one gene member. However, unlike in the study of Zhao et al. [26], there was no S12 subgroup in our gene family classification result. Based on the bidirectional best hits (BBH) between 210 *PtrMYB2R* genes and the *AthR2R3-MYB* genes, no *PtrMYB2R* genes were found to be homologous to the *AthR2R3-MYB* genes belonging to the S12 subgroup. In addition, a total of 21 *R2R3-MYB* genes were clustered together in a single cluster, and they shared high sequence similarity with *AtR2R3-MYB* genes in nine subgroups. However, they were not classified into any of the known subgroups in *A. thaliana* and were thus assigned to the subgroup “Other”.

### 2.5. Collinearity Analysis of R2R3-MYB Genes

Gene duplication was one of the primary driving forces in plant genome evolution. Recent studies have shown that tandem and segmental duplication events are likely to be key factors in the diversification and functional evolution of *R2R3-MYB* genes [37]. Hence, we performed collinearity analysis and discovered that the poplar *R2R3-MYB* gene family had two major gene expansion patterns: WGD/segmental duplication events and tandem duplication (Appendix A). Twelve pairs of tandem duplication events were detected, and these clusters of tandem duplication genes were mainly located on the Chr01, Chr03, Chr11, Chr13, Chr17, and Chr19 chromosomes, involving 19 genes in the S02, S04, S06, and S09 subgroups (Figure 5). A total of 171 *PtrMYB2R* genes (81.4%) were implicated in the WGD/segmental duplication event, which was mainly caused by whole-genome replication. These results suggested that the evolutionary development of the poplar *R2R3-MYB* gene family was influenced by both tandem duplication events and WGD/segmental duplication events, in which WGD might play a crucial role. WGD events certainly contributed to the expansion and functional diversification of *R2R3-MYB* genes in the *P. trichocarpa* genome, providing adequate preparation for angiosperms in response to drastic environmental changes [10]. A total of 156 duplicated gene pairs in the *R2R3-MYB* family were identified (Figure 5 and Appendix A), and nonsynonymous (Ka)/synonymous (Ks) analyses of these gene pairs were subsequently performed. The results showed that the Ka/Ks ratios of all duplicate gene pairs were less than 1 (Appendix A), indicating that the poplar *R2R3-MYB* gene family might be subject to purifying selection during genome evolution [38]. In addition, the Ks value is proportional to the occurrence time of the gene duplication events, which can be used to estimate the divergence time of duplicated gene pairs [39]. Therefore, we analyzed their Ks divergence time and found that 48% of the homologous gene pairs were produced between 8 and 31 million years ago (Mya), which was after the separation of *Populus* and *Salix* (45 Mya) [40]. Approximately 24% of the duplicated gene pairs were generated after the divergence of *Populus* and *Arabidopsis* lineages (100–120 Mya) [5]. This indicated that a large number of WGD and tandem duplication events occurred after the differentiation of *Populus* with *Salix* and *Arabidopsis*, which led to the rapid expansion of the *R2R3-MYB* gene family in *Populus* species.

To understand the phylogenetic relationship of *R2R3-MYB* genes between poplar and the other four species, we performed an interspecific collinearity analysis (Figure 6). The four species consisted of two *Salix* species (*S. purpurea* and *S. suchowensis*), *A. thaliana*, and *O. sativa*. The number of orthologous *R2R3-MYB* pairs of *P. trichocarpa* with *S. purpurea*, *S. suchowensis*, *A. thaliana*, and *O. sativa* were 276, 266, 89, and 53, respectively. It can be seen that there was stronger collinearity between *P. trichocarpa* and the two *Salix* species (*S. purpurea* and *S. suchowensis*), followed by *A. thaliana* and finally *O. sativa*. This phenomenon could be because the *Salix* and *Populus* genera were a sister group in the family Salicaceae. The species belonging to the *Salix* and *Populus* genera shared a relatively close evolutionary relationship and a high level of sequence similarity within their *R2R3-MYB* genes. It seemed obvious that a large proportion of the orthologous *R2R3-MYB* pairs between *P. trichocarpa* and the two *Salix* species were located on the collinear blocks between the genomes of *P. trichocarpa* and the two *Salix* species. This finding also implied that some of the *R2R3-MYB* genes in the *P. trichocarpa* genome suffered from WGD/segmental duplication events during their evolution.

### 2.6. Poplar R2R3-MYB Gene RNA-Seq Analysis

To select putative drought-responsive *R2R3-MYB* genes in *P. trichocarpa*, we reanalyzed RNA-seq profiling from the NCBI SRA database of *PtrMYB2Rs* in three different tissues under drought treatment. And then we selected 10 *R2R3-MYB* genes for further expression pattern analysis. As shown in Figure 7A, *PtrMYB2R028* and *PtrMYB2R151* were highly expressed in all tissues (log_2_(FPKM + 1) ≥ 7), and *PtrMYB2R060* was moderately expressed in all tissues (7 > log_2_(FPKM + 1) ≥ 5). Some genes showed tissue-specific expression. For example, *PtrMYB2R176* was only highly expressed in leaves, *PtrMYB2R089* was preferentially expressed in stems, and *PtrMYB2R111* and *PtrMYB2R119* were specifically expressed in roots.

We also compared the transcript abundance of these *PtrMYB2R* genes under normal conditions and drought stress (short-term and long-term drought stress). Under long-term drought stress, the expression levels of most *PtrMYB2R* genes were significantly changed (*p* < 0.05) and most of them were upregulated, with the exception of *PtrMYB2R033* and *PtrMYB2R060*. In contrast, the expression levels of most genes did not change significantly after short-term drought treatment (*p* > 0.05). The results indicated that *PtrMYB2R* genes might play an important regulatory role in response to drought stress, especially when plants are faced with prolonged drought.

### 2.7. PdMYB2R Gene Cloning and the Expression Patterns of PdMYB2R Genes in Different Tissues

We cloned and sequenced these 10 *R2R3-MYB* genes from the poplar genotype NL895 (*P. deltoides* × *P. euramericana* cv. Nanlin895). They were named according to their corresponding *PtrMYB2R* gene names (*PdMYB2R028*, *PdMYB2R032*, *PdMYB2R033*, *PdMYB2R060*, *PdMYB2R089*, *PdMYB2R111*, *PdMYB2R119*, *PdMYB2R123*, *PdMYB2R151*, and *PdMYB2R176*). The ORFs (open reading frames) of these genes ranged from roughly 700 to 1400 bp in length (Appendix A). The sequences of these cloned *PdMYB2R* genes had an average similarity of more than 99% with their corresponding *PtrMYB2R* genes.

We then analyzed the expression levels of these *PdMYB2R* genes among three tissues (roots, stems and leaves) of NL895 using qRT-PR. As shown in Appendix A, these genes showed multiple expression patterns among tissues and had certain tissue specificity, which was similar to the results of the RNA-seq data analysis. For example, *PdMYB2R032*/*089*/*123*/*151* genes had similar expression patterns, and their expression levels were significantly higher in roots than in stems and leaves. Among them, the expression level of *PdMYB2R089* in roots was particularly significant compared to that in leaves and stems, and its expression level was 290 times higher than that in leaves. The expression levels of *PdMYB2R111*/*119* in leaves were approximately 1.5–3 times those of the other tissues. *PdMYB2R176* had a similar expression pattern, but it is worth noting that the expression level of *PdMYB2R176* in leaves was approximately 25.5 times that in roots and approximately 2.7 times that in stems. However, *PdMYB2R033* had a significantly higher expression level in stems than in the other two tissues. The remaining two genes, *PdMYB2R028* and *PdMYB2R060*, were not significantly different between the three tissues.

### 2.8. Analysis of the Expression Pattern of Poplar PdMYB2R Genes under Drought Stress

Drought is one of the important environmental factors restricting the growth and development of woody plants, and even severe drought may result in whole-plant mortality [41]. Hence, we further explored the drought-induced expression patterns of *PdMYB2R* genes in three different tissues using qRT-PCR. The change trends of different genes at different drought treatment time points were shown in Figure 7B. Nine genes excluding *PdMYB2R119* showed similar patterns in leaves, all exhibiting a trend of first increasing and then decreasing. The relative expression levels of most genes peaked at 6 h or 12 h after PEG6000 treatment and then decreased to the initial expression levels. The expression patterns of *PdMYB2R* genes in roots and stems fluctuated at individual time points. For example, *PdMYB2R032*, *PdMYB2R033*, *PdMYB2R060*, *PdMYB2R123*, and *PdMYB2R176* in the roots and/or stems showed a trend of slight decrease at 1 h, rapid increase to peak value at 6 h, and then rapid decline. In addition, at the late stage of drought treatment (3 d or 5 d), individual genes were slightly upregulated in some tissues, such as *PdMYB2R033*, *PdMYB2R060* (roots and stems), and *PdMYB2R111* (roots), but the changes were not significant and did not affect the overall trend.

Under drought conditions, 10 genes were induced to varying degrees in roots, stems, and leaves. Most genes (except *PdMYB2R119*) had higher expression levels in leaves than in roots and stems. Especially when poplar was subjected to drought stress for 6 h, the expression levels of *PdMYB2R060*, *PdMYB2R111*, and *PdMYB2R123* in leaves were significantly upregulated and were respectively approximately 100, 310, and 500 times higher than those under normal conditions. It is worth noting that *PdMYB2R119* was preferentially upregulated in response to instant drought treatment in roots but significantly downregulated in stems and leaves under 2–5 days of drought treatment. Similarly, *PdMYB2R032* was also significantly downregulated in leaves and stems under 2–5 days of drought stress, suggesting that *PdMYB2R032* and *PdMYB2R119* might be tissue-specific in different stress periods. In addition, we discovered that all *PdMYB2R* genes had similar change trends in the two tissues, and there were no genes with completely identical change trends among the three tissues, which further validated that *PdMYB2R* has certain tissue specificity.

Our results demonstrated that the relative expression levels of most genes changed significantly under short-term drought stress (within 1 day), but the changes were not obvious under 1- to 5-day PEG6000 treatment, and the expression levels of most *PdMYB2R* genes tended to be stable at 1 day after exposure to the drought treatment. There was a slight difference from the *P. trichocarpa* RNA-seq data, which might be attributed to the different methods of drought treatment and plant materials. In this study, the longest stress time was 5 days, so there was a certain uncertainty about the longer stress time point. In conclusion, the transcriptome analysis of *P. trichocarpa* and the qPCR results of *PdMYB2R* showed that the expression levels of these 10 *R2R3-MYB* genes all changed significantly under the drought environment. These results indicated that these *R2R3-MYB* genes might play an important role in *Populus* species under drought stress. However, their specific regulatory patterns and mechanisms need to be further investigated.

### 2.9. Subcellular Localization and Transcriptional Activation Activity

The subcellular location of R2R3-MYB TFs might be involved in their regulatory role in the transcriptional regulatory network. Thus, we used three online software tools to predict the subcellular localization of poplar R2R3-MYB protein. The results showed that only PtrMYB2R041 (Potri.003G123800.1) and PtrMYB2R124 (Potri.010G240800.1) of all 210 PtrMYB2R TFs were predicted to be located in the extracellular and cytoplasmic regions. The remaining 208 R2R3-MYB proteins were predicted to have a nuclear localization signal (Appendix A). Then, we randomly selected two putative nuclear-located *R2R3-MYB* genes for validating subcellular localization. The subcellular localization assay showed that the encoded proteins of *PdMYB2R032* and *PdMYB2R151* only exhibited fluorescent signal in the nucleus but not in the cytoplasm and cell membrane (Figure 8A). These findings indicated that the two R2R3-MYB TFs might function in the nucleus of *Populus* species.

Then, we investigated the autoactivation activities of the two TFs by transforming yeast. The two fusion plasmids, including pGBKT7-PdMYB2R032 and pGBKT7-PdMYB2R151, were transformed into yeast cells AH109. The results of transcriptional activation activity showed that both PdMYB2R032 and PdMYB2R151 proteins could grow on SD/Trp medium as well as negative and positive controls, indicating that they were nontoxic. As shown in Figure 8B, only the transformants containing pGBKT7-35 can grow on the triple deficient medium (SD/-Trp/-Ade/-His), and showed blue after adding X-α-Gal. However, pGBKT7, pGBKT7-PdMYB2R032 and pGBKT7-PdMYB2R151 could not grow on the triple deficient medium, indicating that the full-length sequences of PdMYB2R032 and PdMYB2R151 did not possess transcriptional activation activity for activating the expression of downstream reporter genes.

## 3. Discussion

### 3.1. Identification and Evolution of the Poplar R2R3-MYB Gene Family

To the best of our knowledge, the identification of R2R3-MYB TFs in the entire *P. trichocarpa* genome (v3.0) has been reported in four papers. However, some differences persisted in the identification results of the previously published papers. These differences were mainly due to the diversity and complexity of the conserved domain of *MYB* genes, resulting in unclear or incorrect gene classification of *MYB* subfamilies [11]. In the present study, we identified a total of 210 *PtrMYB2R* gene members in *P. trichocarpa*. The poplar *R2R3-MYB* gene family had a larger number of gene members than other identified dicotyledonous families, such as *A. thaliana* (126) [42], *Eucalyptus grandis* (141) [21], *Malus domestica* (186) [43], *Cinnamomum camphora* (96) [44], and *Camellia sinensis* (112) [45]. The *R2R3-MYB* gene family may have undergone functional diversification during genome evolution in plants, thus forming different subfamilies [11]. In this study, the poplar *R2R3-MYB* gene family was phylogenetically clustered into 23 subgroups. Although the functions of most *PtrMYB2R* genes have not been characterized, 90% of the *PtrMYB2R* genes are clustered in the functional groups of *Arabidopsis*. Many *PtrMYB2R* genes were grouped into subgroups S13, S15, and S22, the members of which are involved in abiotic stress responses in *Arabidopsis*. Consequently, phylogenetic analysis will aid in the identification of *PtrMYB2R* genes related to the drought stress response [46].

Many studies have shown that whole-genome duplication (WGD) and fragment duplication events are important mechanisms by which the *R2R3-MYB* gene family has rapidly expanded during poplar genome evolution [28]. Chromosomal mapping analysis showed that 207 *R2R3-MYB* genes were unevenly distributed over 19 *P. trichocarpa* chromosomes, and most *R2R3-MYB* genes were concentrated on certain chromosomal regions. This finding was similar to the uneven chromosomal distribution of poplar *MYB-related* genes, which might be caused by multiple DNA duplications, such as WGD, tandem duplication and dispersed duplication [47]. Then, we found a large number of WGD events and small-scale tandem duplication events among the poplar *R2R3-MYB* gene family, and most paralogous gene pairs diverged after the separation of *Populus* and *Salix* (45 Mya). The results showed that WGD events played a leading role in the rapid expansion of the poplar *R2R3-MYB* gene family, and small tandem repeat events also played a role in promoting the expansion [11]. The poplar *R2R3-MYB* family was also likely to suffer rapid expansion after its divergence with monocotyledons and *Salix* species. In addition, we discovered that the *R2R3-MYB* duplicated gene pairs of poplar might have been mainly affected by purifying selection (Ka/Ks < 1) during evolution, which was consistent with the results obtained for other plants, such as *Liriodendron chinense* and *Solanum tuberosum* [48,49]. These duplicated *R2R3-MYB* genes might be associated with the gain of new functions that increase ecological resilience to a variety of adverse environmental stresses [48].

### 3.2. Analysis of Functional Sites of Poplar R2R3-MYB Proteins

Understanding the spatial information of the proteins encoded by the *R2R3-MYB* genes in cells can provide a reference for judging its protein function. The subcellular localization prediction results showed that 205 of the 210 (97.6%) PtrR2R3-MYB proteins might be localized in the nucleus, which is consistent with the R2R3-MYB protein research of other species [45,50]. The subcellular localization experiment results confirmed our prediction, indicating that they were nuclear-located proteins, which was consistent with the theory that TFs normally functioned in the nucleus [51,52]. TFs can regulate target genes by binding specifically to the *cis*-element of their promoters in the nucleus [53]. R2R3-MYB TFs could bind specifically to the *cis*-acting elements of drought-responsive genes, resulting in control of their expression levels [7].

This study showed that PdMYB2R032 and PdMYB2R151 transcription factors have no non-autoactivation activity and can be directly used for subsequent protein interaction experiments with drought-related proteins [54]. These results suggest that these R2R3-MYB proteins may interact with other proteins to form transcription complexes to regulate downstream gene expression [55,56,57]. The regulatory relationship between PdMYB2R032/151 and other cofactors or key downstream drought-related genes can be further verified by EMSA (electrophoresis mobility shift assay), ChIP (chromatin immunoprecipitation), and yeast one-hybrid or two-hybrid technologies in the future [56,58].

### 3.3. Tissue-Specific Expression Pattern of Poplar R2R3-MYB Genes under Drought Stress

Research related to the function of the R2R3-MYB transcription factor has mainly focused on the herbaceous model species *A. thaliana*, and approximately 80% of functional research on *AtR2R3-MYB* genes has been reported [11,19,59,60]. Only a few drought-related *R2R3-MYB* genes, such as *PtoMYB142* and *PtrMYB94*, have been reported in poplar as a woody model species [16,18]. Furthermore, the study characterized the drought-inducible expression patterns of 10 *PdMYB2R* genes across different tissues in poplar NL895. Despite some differences between the qRT-PR results and *P. trichocarpa* RNA-seq profiling, the expression levels of these 10 *R2R3-MYB* genes were significantly changed in three poplar tissues (roots, stems and leaves) in response to drought stress. These *PtrMYB2R* genes shared a relatively similar expression pattern, but their expression levels differed significantly among the three tissues. This result suggested that these *PtrMYB2R* genes might play similar regulatory roles under drought stress. The expression of genes is determined by their promoters in plants, and we identified a large number of drought-related elements (e.g., MYB, MYC, ABRE, TATA-box, G-box, W-box, and MBS) in gene promoter regions. In conclusion, *PtrMYB2R* genes are generally involved in abiotic stress such as drought.

We found that some orthologues of *R2R3-MYB* genes in *Arabidopsis* may have similar regulatory roles in poplar under drought stress [26]. For example, Liang et al. (2005) found that overexpression of *AtMYB61* (AT1G09540) resulted in decreasing stomatal pore size and affected gas exchange, which could improve the water-use efficiency of plants in water-deficient environments [61]. Homologous genes of *AtMYB61*, *PdMYB2R033* and *PdMYB2R060* showed significant changes after drought treatment. This result implicated their potential function in contributing to plant tolerance under drought stress. A large number of abiotic stress-related *cis*-elements were found in the promoter regions of four genes (*AtMYB44*, *AtMYB73*, *AtMYB77*, and *AtMYB70*) in the *Arabidopsis R2R3-MYB* family S22 subgroup. The overexpression of *AtMYB44* could enhance the ability of *Arabidopsis* to resist drought, and loss of the *AtMYB73* causes hyper-induction of the *SOS1* and *SOS3* genes in response to high salinity [14,62]. Three poplar *R2R3-MYB* genes, *PdMYB2R028*, *PdMYB2R111*, and *PdMYB2R151*, belonged to the *R2R3-MYB* family subgroup S22 and were also homologous to *AtMYB73*. The three genes were significantly induced in three poplar organs under drought stress. The results suggested that these *R2R3-MYB* genes belonging to subgroup S22 might be implicated in the drought-responsive processes of poplar.

The regulatory roles of some poplar *R2R3-MYB* genes in drought response have been reported. For example, recent studies have demonstrated that *PtrMYB121* (Potri.002G185900) and *PtoMYB170* (Potri.005G001600) can promote the accumulation of lignin and cellulose, and *PtoMYB170* enhances drought tolerance by triggering stomatal closure [17,63]. As a transcriptional activator, *PtrMYB94* (Potri.017G082500) was involved in the transcriptional regulation of the expression of ABA and drought stress-related genes (e.g., *ABA1*, *DREB2B*, and *P5CS2*), thereby improving the tolerance of transgenic *A. thaliana* plants to drought stress [16]. Similar to the results of Fang et al., our qRT-PR results showed that the expression level of *PdMYB2R176* (Potri.017G082500) had been increased four times in the leaves of the poplar genotype NL895 at 6 h after PEG6000 treatment compared with the control group (0 h). Taken together, the data suggested that these *R2R3-MYB* gene members might play a crucial role in drought stress response.

## 4. Materials and Methods

### 4.1. Genomic Data Retrieval

The genome data of *P. trichocarpa* (v3.0) and *Salix purpurea* (v5.1), including genomic DNA sequences, proteins, protein-encoding genes and genome-annotation files (GFF/GTF, general feature format/gene transfer format), were downloaded from the plant genome database Phytozome (https://phytozome.jgi.doe.gov/, accessed on 5 March 2022). We also obtained the genomic information file of *Salix suchowensis* (GCA_017552425.1) from the NCBI genome database (https://www.ncbi.nlm.nih.gov/genome/, accessed on 5 March 2022). In addition, the *R2R3-MYB* family relevant genomic data of *A. thaliana* (TAIR10) and *Oryza sativa* subsp. japonica (v1.0) were downloaded from the Ensembl Plants website (release 53, https://plants.ensembl.org/info/data/ftp/index.html, accessed on 5 March 2022).

### 4.2. Identification of R2R3-MYB Transcription Factors in Poplar

First, the hidden Markov model (HMM) file of the MYB binding domain (Pfam no. PF00249.31) was retrieved from the Pfam database website (https://pfam.xfam.org/, accessed on 2 May 2022) and used to search MYB repeats in the *P. trichocarpa* proteins using hmmsearch in the HMMER package (v3.3.2, http://hmmer.org/, accessed on 2 May 2022) with an E-value cutoff of 1 × 10^−3^. Moreover, the candidate *P. trichocarpa* R2R3-MYB (PtrMYB2R) proteins were inspected using BLAST+ (v2.9.0) against *A. thaliana* R2R3-MYB (AthR2R3-MYB) proteins obtained from the *Arabidopsis* Information Resource (TAIR, https://www.arabidopsis.org/, accessed on 2 May 2022). The BLASTp hits with an E-value of less than 1 × 10^−5^ were retained. The proteins obtained by the above two methods are potential *R2R3-MYB* gene family members. To further confirm the reliability of candidate R2R3-MYB proteins, potential PtrMYB2R sequences were submitted to Pfam (http://pfam.xfam.org/, accessed on 2 May 2022), CDD (https://www.ncbi.nlm.nih.gov/cdd/, accessed on 2 May 2022) and SMART (http://smart.embl-heidelberg.de/, accessed on 2 May 2022) to check the sequences of intact R2 and R3 MYB repeats and to ensure the completeness of the MYB domain. Potential proteins with incomplete conserved MYB domains were eliminated, and the remaining proteins were considered to belong to the poplar R2R3-MYB family.

### 4.3. Chromosomal Location, Physicochemical Properties and Cis-Acting Element Analysis of R2R3-MYB Genes in Poplar

The chromosomal distribution information of *PtrMYB2R* genes was extracted from the *P. trichocarpa* genome annotation file in GFF/GTF format. MapInspect software was used to map the location of *R2R3-MYB* genes on poplar chromosomes, and they were named according to their corresponding locations on all chromosomes. Then, the physicochemical properties of these PtrMYB2R proteins, such as the isoelectric point (pI), molecular weight (MW), protein length, instability index, aliphatic index and hydropathicity, were calculated using the Python package Biopython (v1.79, https://biopython.org/, accessed on 5 May 2022) and the ProtParam tool on the Expasy website (https://www.expasy.org/resources/protparam, accessed on 5 May 2022). Finally, we predicted the *cis*-acting elements in the upstream 2,000-bp region of the poplar *R2R3-MYB* genes using the online software PlantCARE (http://bioinformatics.psb.ugent.be/webtools/plantcare/html/, accessed on 5 May 2022).

### 4.4. Multiple Sequence Alignment and Phylogenetic Analysis

Multiple sequence alignment (MSA) of PtrMYB2R proteins was carried out using ClustalW (v2.1, http://www.clustal.org/clustal2/, accessed on 5 May 2022) under default parameter settings. The protein substitution model (JTT+F+R8) with the minimum BIC (Bayesian information criterion) score was inferred from the PtrMYB2R MSA files by ModelFinder in IQ-TREE (v1.6.12) [64]. The ML (maximum likelihood) phylogenetic tree was reconstructed using IQ-TREE (v1.6.12) with the optimal substitution model JTT+F+R8 and 1000 bootstrap replicates [65]. The resulting ML tree was visualized using the R package ggtree (v3.2.1) [66].

### 4.5. PtrMYB2R Gene Expression Analysis

The RNA-seq data of poplar roots, stems and leaves at three-time points were downloaded from the NCBI SRA database (https://www.ncbi.nlm.nih.gov/sra, accessed on 5 May 2022) under BioProject no. PRJEB19784 [67]. The three-time points were composed of an untreated control point, short-term drought (5 days after withholding water), and prolonged drought (5 days after withholding water and 7 days after limited watering). RNA-seq data were checked for quality control using FastQC (v0.11.9), and reads were trimmed with Trimmomatic (v.0.38) [68]. Then, the RNA-seq reads were mapped against the *P. trichocarpa* (v3.0) genome using the spliced aligner STAR (v2.7.3a) software [69]. The featuresCounts (v2.0.1) software was used to count the numbers of reads that were aligned on the *P. trichocarpa* genome [70]. Finally, the expression patterns of *PtrMYB2Rs* in three tissues and under three treatment time points were analyzed with the R package DESeq2 (v1.30.1) [71].

### 4.6. Genome Collinearity Analysis

The collinearity relationship between *R2R3-MYB* genes in the *P. trichocarpa* genome poplar was visualized with the R package circlize (v0.4.15) [72]. To further explore the evolutionary mechanism of the poplar *R2R3-MYB* gene family, interspecific collinearity analysis was performed on five flowering plants, including *P. trichocarpa*, *Salix purpurea*, *S. suchowensis*, *A. thaliana*, and *Orzya sativa*. MCScanX (https://github.com/wyp1125/MCScanX, accessed on 6 May 2022) and BLASTP (E-value threshold of 1 × 10^−10^) were used to identify gene tandem duplication and fragment duplication [73]. The collinear relationship between poplar and the other four species was then visualized using the Python package JCVI (v0.8.12, https://github.com/tanghaibao/jcvi, accessed on 6 May 2022) [74]. Paralogous and orthologous gene pairs were aligned with ClustalW (v2.1), and nonsynonymous (Ka)/synonymous (Ks) for each gene pair was calculated using ParaAT (v2.0) and KaKs-Calculator (v2.0) [75]. The divergence time of duplicated gene pairs was calculated with their Ks values and the synonymous mutation rate (r = 9.1 × 10^−9^) [76,77].

### 4.7. Cloning of R2R3-MYB Genes from NL895 Poplar

In this study, the tissue-cultured plantlets of hybrid poplar NL895 (*P. deltoides* × *P. euramericana* cv. Nanlin895) were used for cloning 10 selected *R2R3-MYB* genes. These plantlets were grown on 1/2 Murashige and Skoog (MS) medium under a photoperiod of 16 h of light and 8 h of darkness. The young leaves were harvested from 45-day-old plantlets, frozen immediately in liquid nitrogen, and stored at −80 °C for further RNA isolation.

Total RNA was extracted from poplar NL895 leaves using an RNA extraction kit (Tiangen, Beijing, China), and 1 μg of RNA was reverse-transcribed into cDNA. Then, the high-fidelity PCR enzyme KOD-Plus-Neo (Toyobo, Osaka, Japan) was used for PCR amplification, and the cloning primers in the reaction system are shown in Appendix A. The PCR product was verified by 1% agarose gel electrophoresis, and the single and correct band was purified and recovered. The recovered products were connected to pTOP001 Blunt Simple Cloning vector (Genesand, Beijing, China), transformed into *E. coli* competent cells TSC-C01 (Tsingke, Beijing, China) and cultured overnight at 37 °C. The next day, the monoclonal bacteria were selected for amplification and sent to the biotechnology company (Sangon Biotech, Shanghai, China) for Sanger sequencing.

### 4.8. Expression Pattern of R2R3 MYB Genes in Poplar NL895 under Drought Stress

The tissue-cultured plantlets of NL895 were grown at an air temperature of 25 °C and 60% relative humidity. NL895 plantlets 45 days old were subjected to water-deficit stress imposed by polyethylene glycol 6000 (PEG6000). The NL895 plantlets with consistent growth conditions were placed in solid 1/2 MS medium containing 10% PEG6000 for 0 h, 1 h, 6 h, 12 h, 24 h (1 day), 72 h (3 days), and 120 h (5 days). Three biological replicates were performed separately for each time point, and each biological replicate was composed of 3~5 leaves. As previously described, total RNA was isolated from the leaf, root and stem tissues of the NL895 plantlets under PEG6000 treatment, and 1 μg of RNA was reverse-transcribed into cDNA. Real-time quantitative PCR (qPCR) was used to detect the expression pattern of these 10 poplar *R2R3-MYB* genes under drought stress. These qPCR data were normalized and analyzed using the 2^-ΔΔCt^ algorithm [78] with the poplar *EF1-α* gene (NCBI GenBank no. GQ253565.1) as an internal reference gene. Using the cloned *PdMYB2R* gene coding sequence (CDS) as the template, qPCR primers were designed on the website of GenScript (https://www.genscript.com/, accessed on 6 June 2022). The primer sequences are shown in Appendix A.

### 4.9. Subcellular Localization of the R2R3-MYB Transcription Factor in Poplar

The subcellular localization of PMYB2R proteins was predicted using three online tools, including BUSCA (http://busca.biocomp.unibo.it/, accessed on 20 June 2022), Cello (http://cello.life.nctu.edu.tw/) and WoLFPSORT (https://wolfpsort.hgc.jp/, accessed on 20 June 2022). To ensure the reliability of the prediction results, we combined the three prediction results to determine the final subcellular localization of each *R2R3-MYB* gene.

The PEG-mediated transformation of poplar NL895 protoplast was performed according to the method described by Tan et al. [79]. Two poplar R2R3-MYB proteins were selected as the target genes to verify the subcellular localization. Specifically, the CDS of PdMYB2R032 and PdMYB2R151 without a stop codon was inserted into the vector p2GWF7.0 for fusion expression of the target gene and green fluorescent protein (GFP). D53-mCherry, a nuclear localization protein with a red fluorescent label, was used as a positive control [80]. First, poplar leaves were chopped into tiny pieces, and protoplasts were obtained by enzymatic hydrolysis with pectinase (Sigma, Saint Louis, MO, USA) and cellulase (Sigma, Saint Louis, MO, USA). Then, the plasmid containing the target gene was transformed into poplar NL895 protoplasts. Finally, the subcellular localization information of the two target proteins was captured under an AxioScope A1 fluorescence microscope (Zeiss, Oberkochen, Germany).

### 4.10. Transcriptional Activation Assay

The full-length of *PdMYB2R032* and *PdMYB2R151* cDNA were cloned into the pGBKT7 plasmid (VT006, Coolaber, Beijing, China). The two fusion plasmids, including pGBKT7-PdMYB2R032 and pGBKT7-PdMYB2R151, were transformed into yeast cells AH109 (CC300, Coolaber, Beijing, China). In the study, empty pGBKT7 vector and pGBKT7-53 vector (VT007, Coolaber, Beijing, China) were used as negative and positive controls, respectively. Finally, the transformed yeast cells were streaked and cultured on SD medium (SD/-Trp and SD/-Trp/-Ade/-His) at 29 °C for 48 h.

## 5. Conclusions

In this research, a total of 210 members belonging to the *R2R3-MYB* gene family were identified in the entire *P. trichocarpa* nuclear genome and were phylogenetically divided into 23 subgroups. A total of 207 *R2R3-MYB* genes were relatively unevenly distributed on the 19 *P. trichocarpa* chromosomes. A large number of whole-genome duplications and a few tandem duplication events were found in these *R2R3-MYB* genes, suggesting that whole-genome duplication might have made a large contribution to the rapid expansion of the poplar *R2R3-MYB* family. There were a large number of drought stress-related *cis*-acting elements in the promoter regions of *R2R3-MYB* genes, suggesting that the *R2R3-MYB* genes might be involved in the response to drought stress. Almost all poplar R2R3-MYB transcription factors were predicted to be localized in the nucleus, and the nuclear positioning of two R2R3-MYB proteins was validated in a subcellular localization analysis in poplar NL895 protoplasts. In addition, RNA-seq profiling and qRT-PCR analysis showed that the expression levels of 10 *R2R3-MYB* genes changed significantly in different tissues under drought stress. However, how these R2R3-MYB proteins bind to specific cis-acting elements of drought-related target genes to regulate drought response in poplar remains largely unclear. These can be studied in the future by ChIP, EMSA, and yeast one/two-hybrid approaches. This study will lay a foundation for further investigation of the regulatory roles of R2R3-MYB transcription factors in *Populus* species exposed to water deficit, and provide support for the development of new poplar genotypes with elevated drought tolerance.

## Figures and Tables

**Figure 1 ijms-24-05389-f001:**
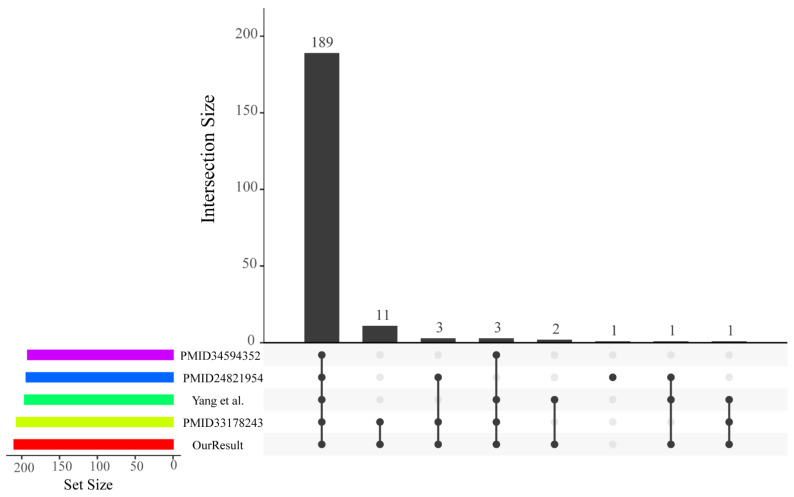
Visualization of intersecting sets in our and four previous genome-wide identifications of *R2R3-MYB* genes in *P. trichocarpa*. Our result for the poplar *R2R3-MYB* family identification is represented by the red horizontal bar. The other four articles—PMID: 34594352 (https://pubmed.ncbi.nlm.nih.gov/, accessed on 1 May 2022) [28], PMID: 24821954 (https://pubmed.ncbi.nlm.nih.gov/, accessed on 1 May 2022) [25], Yang et al. [27], and PMID: 33178243 (https://pubmed.ncbi.nlm.nih.gov/, accessed on 1 May 2022) [26]—are colored in pink, blue, green, and yellow, respectively. The matrix layout shows all intersections of five *R2R3-MYB* identification sets. Black solid points in the matrix denote sets that are intersecting.

**Figure 2 ijms-24-05389-f002:**
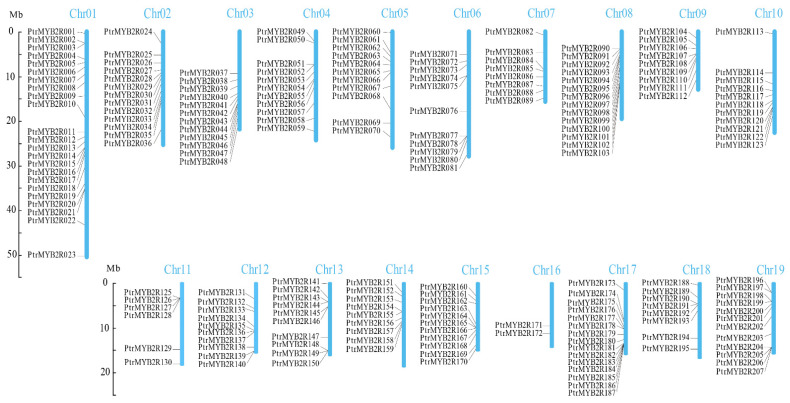
Distribution of *PtrMYB2R* genes on the 19 poplar chromosomes. The 19 chromosomes of *P. trichocarpa* are represented by the blue vertical bars, and their chromosome numbers are at the top of the chromosomal bars. A total of 207 *R2R3-MYB* genes are distributed over the *P. trichocarpa* chromosomes. The chromosomal scale bar is 10 Mb.

**Figure 3 ijms-24-05389-f003:**
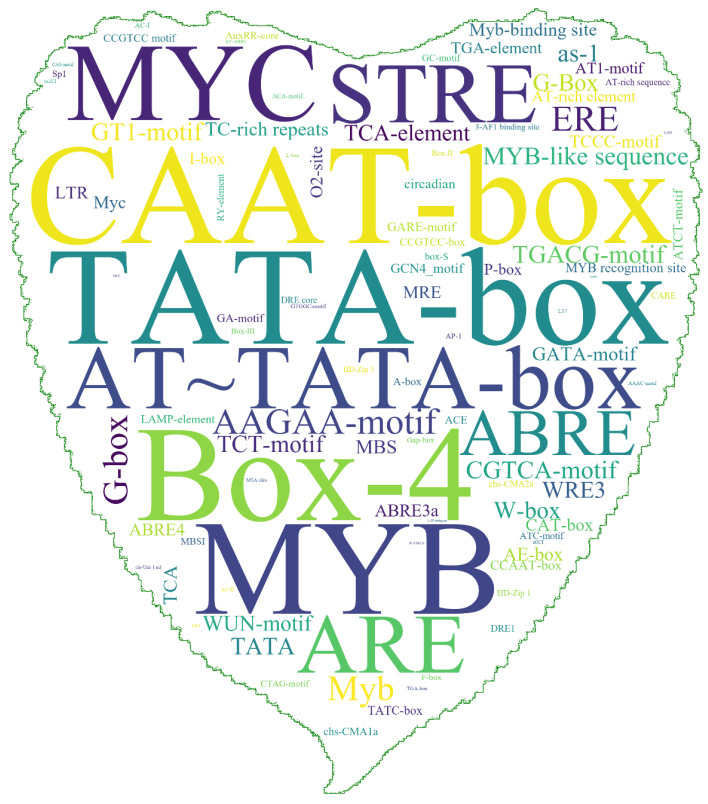
*Cis*-acting elements in the promoter region of all poplar *R2R3-MYB* genes. The size of a word in the word-cloud image indicates the frequency of the *cis*-acting elements within the upstream promoter region of all *R2R3-MYB* genes on the *P. trichocarpa* genome. The picture is the shape of the *P. euramericana* leaf.

**Figure 4 ijms-24-05389-f004:**
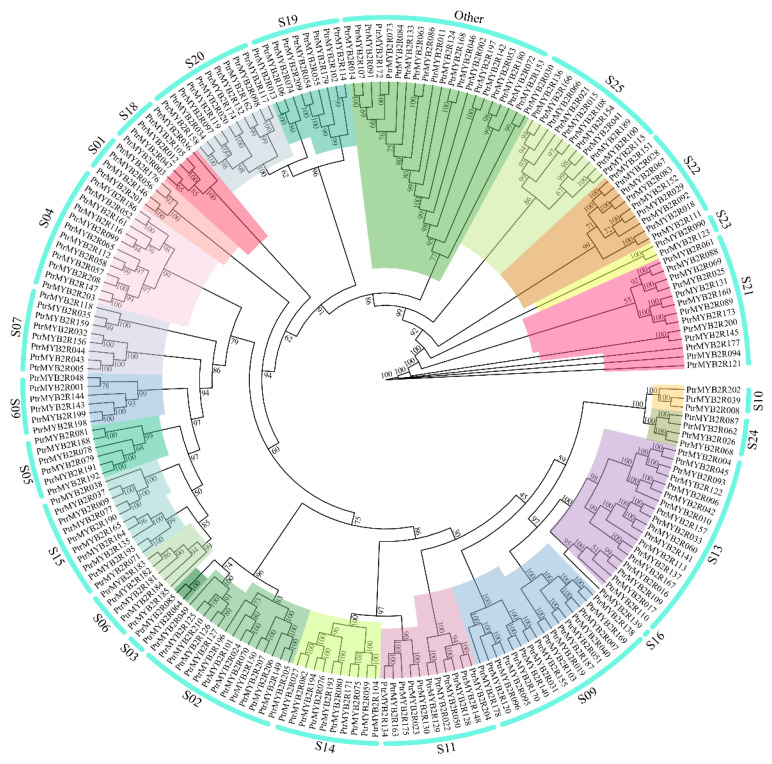
Phylogenetic tree of poplar *R2R3-MYB* gene family. The poplar *R2R3-MYB* family is divided into 23 subfamilies which are marked in different colors. The outer circle shows the numbers of these 23 subgroups. The bootstrap values are shown at branch nodes of the phylogenetic tree.

**Figure 5 ijms-24-05389-f005:**
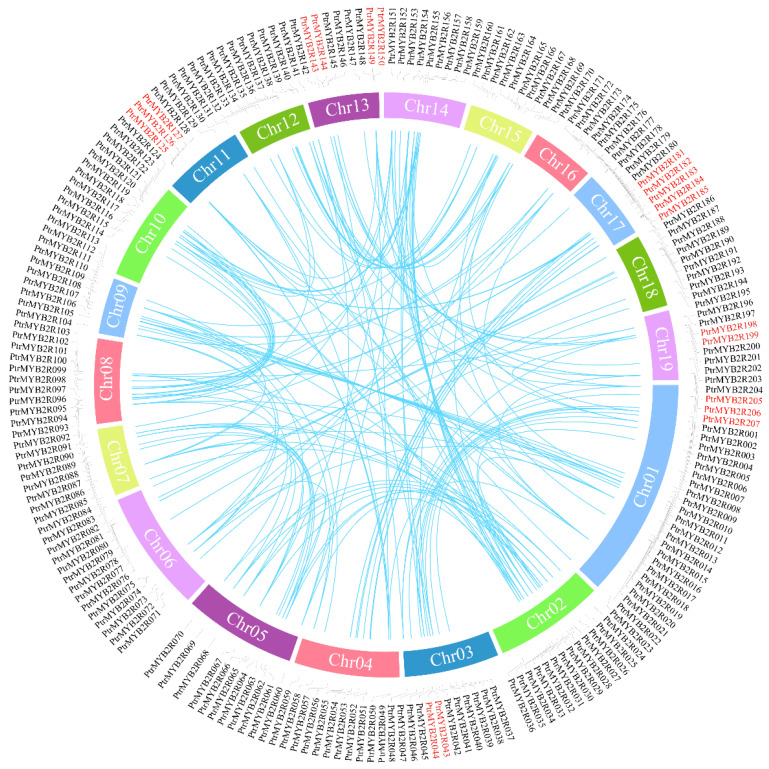
Collinearity analysis of poplar *R2R3-MYB* genes. The outer circle shows the chromosomal distribution of these *R2R3-MYB* genes. The tandem duplicated genes on the outer circle are marked in red. The duplicated gene pairs are indicated by blue lines in the inner ring.

**Figure 6 ijms-24-05389-f006:**
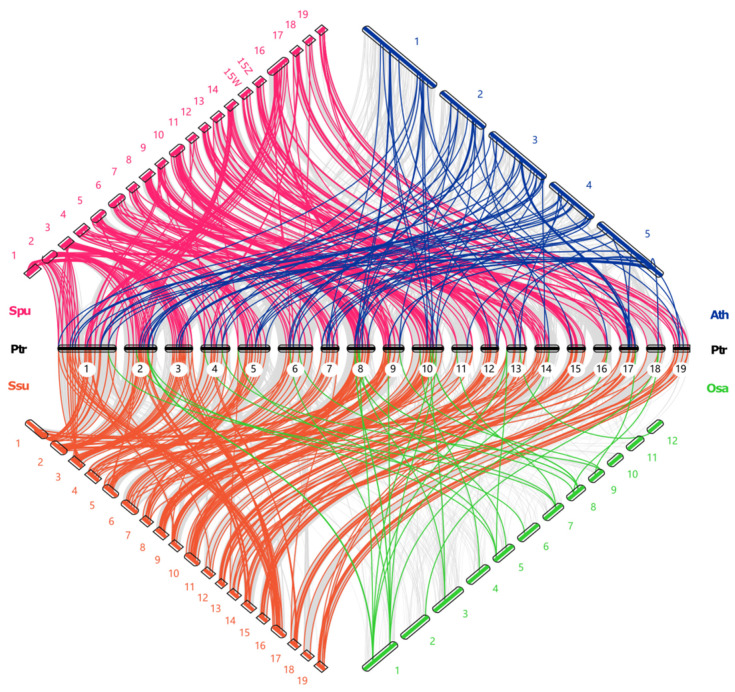
The interspecific collinearity analysis of *R2R3-MYB* genes in poplar and four different species. The chromosomes of the *P. trichocarpa*, *S. purpurea, S. suchowensis*, *A. thaliana* and *O. sativa* genomes are marked in black, pink, orange, blue, and green, respectively. The collinearity blocks between *P. trichocarpa* and other four plant species are marked by gray lines.

**Figure 7 ijms-24-05389-f007:**
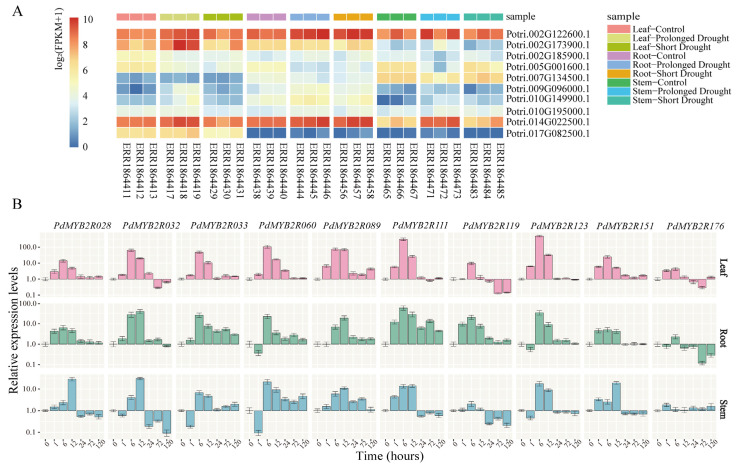
Expression patterns of 10 *R2R3-MYB* genes in poplar. (**A**). RNA-seq expression profiles of 10 *PtrMYB2R* genes in the *P. trichocarpa* under drought stress. The gene ID for the 10 *PtrMYB2R* genes are as follows: *PtrMYB2R028*, *PtrMYB2R032*, *PtrMYB2R033*, *PtrMYB2R060*, *PtrMYB2R089*, *PtrMYB2R111*, *PtrMYB2R119*, *PtrMYB2R123*, *PtrMYB2R151*, and *PtrMYB2R176*. The heatmap is plotted using the log2-transformed (FPKM+1) values. There are three samples per treatment for each tissue, and the sample information is shown on the right side, such as leaf control (SRA accession no. ERR1864411, ERR1864412, and ERR1864413), leaf prolonged drought (ERR1864417, ERR1864418, and ERR1864419), and so on. (**B**). qRT-PCR profiles of 10 *PdMYB2Rs* in NL895 with PEG6000 treatment. The relative expression values of these *PdMYB2R* genes are log_2_-transformed. The leaves, stems, and roots are harvested at 0, 1, 6, 12, 24, 72, and 120 h after 10% PEG6000 treatment.

**Figure 8 ijms-24-05389-f008:**
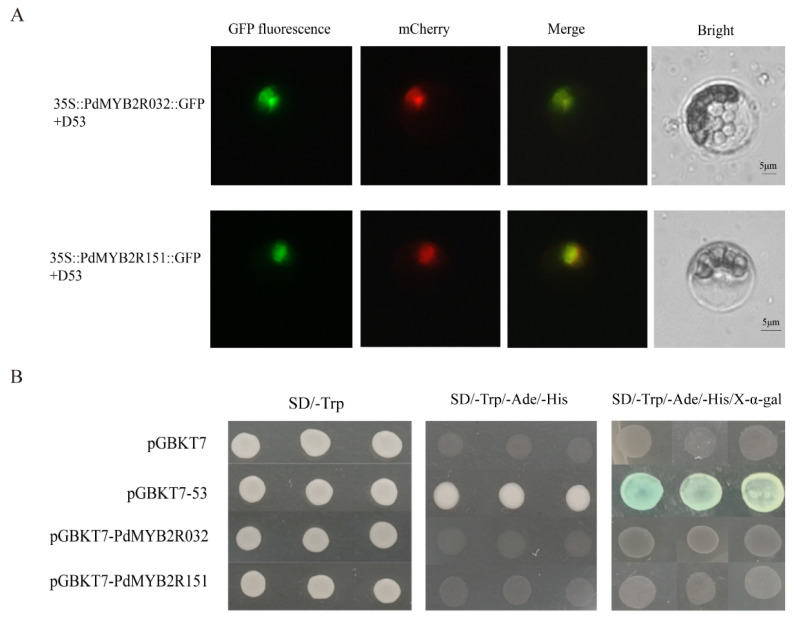
Subcellular localization and autoactivation activity of PdMYB2R032 and PdMYB2R151 proteins. (**A**) Subcellular localization. mCherry-D53, a red and nuclear-localized marker. Scale bar: 5 μm. (**B**) Analysis of autoactivation activity. pGBKT7 vector was the negative control, and pGBKT7-53 vector was the positive control. Yeast cells were grown on SD medium at 29 °C for 48 h.

## Data Availability

All datasets presented in this study are included in the article/Appendix A.

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
