# Peer review of "Comprehensive Genome-Wide Analyses of Poplar R2R3-MYB Transcription Factors and Tissue-Specific Expression Patterns under Drought Stress"

_ijms, 2023, doi:10.3390/ijms24065389_

Round 1
Reviewer 1 Report
I have read a manuscript entitled “Comprehensive genome-wide analyses of poplar R2R3-MYB transcription factors and tissue-specific expression patterns under drought stress.” It describes the authors finding that there are 210 R2R3-MYB transcription factors, more than previous thought, and by comparing their findings with four others side-by-side, the authors show not only where the R2R3-MYB genes intersect four previous genome-wide identification the genes in P. trichocarpa , but that the poplar R2R3-MYB genes underwent rapid expansion. The authors provide a roadmap for where start looking for drought-responsive R2R3-MYB genes in this plant.
The manuscript is well written with very minor corrections needed (below). It is easy to understand in that the results are clearly given with adequate statistic/validation methods, and the conclusions supported by the results. The introduction provide sufficient background and include all references relevant to the research. The study design itself is appropriate for the research. I therefore see no reason of why it should not be published in its current form.
Line 65: Consider replacing “to be” with “being”
Line 69: Consider inserting “also” between “has” and “been”
In the CONCLUSION section, I wonder if some of the tools we have developed could be used in plants, being a molecular and developmental biologist myself, modeling gene regulatory network using the sea urchin (Ref: https://doi.org/10.1006/dbio.2002.0635 ; http://dx.doi.org/10.1387/ijdb.170194oo ; https://doi.org/10.1016/j.ydbio.2004.05.033 ). Perhaps the authors might find one to include here (CONCLUSION section) as a suggestion for future studies, if any.
Reviewer 2 Report
The R2R3-MYB family, one of the largest transcription factor families in higher plants, controls a wide variety of plant-specific processes including, notably, reaction to diferent abiotic stress including drowt tolerance. Poplar is not only a model object, but also a widely cultivated woody plant, and the study of all aspects of its drought tolerance mechanism is undoubtedly important.
The article is multifaceted is extremely relevant and is of undoubted practical and scientific interest. It is undoubtedly a high-level work and the authors obtained significant very interesting results. The article is very interesting, original, the content of the article corresponds to the abstract and title. The tables and figures are complement the text well.
There are some comments and suggestions for authors.
1. In the "Introduction" it is desirable for the authors to more clearly and briefly formulate the aim of the work. This will improve the perception of the article by readers.
2. In the "results" it is better to clearly distinguish between the results obtained in silico and in the experiment by the authors. It seems to me, that it would be appropriate to insert at least a sentence at the beginning of the description of each stage of research, explaining the origin of the analyzed material.
3. Figures 2 and 7 are very hard to read (small print). Perhaps it is better to enlarge figures 2 and 7 and transfer them to supplementary material, and duplicate figure 7 with a table.
4. Line 69 "Genome-wide identification of the R2R3-MYB genes 68 in the Populus trichocarpa genome has been reported in four publications". Are you sure there are only four articles? It seems to me that it is better to rephrase this sentence.
The authors have undoubtedly obtained important results, and I believe that the article can be published after revision.
Reviewer 3 Report
Please see attached
